

# Evaluation of floods in Western/Central Europe: the role of index design

Blanka Gvoždíková[1], Miloslav Müller[1,2]

[1]Faculty of Science, Charles University, Prague, Czech Republic
[2]Institute of Atmospheric Physics AS CR, Prague, Czech Republic

*Correspondence to*: Blanka Gvoždíková (blanka.gvozdikova@natur.cuni.cz)

**Abstract.** This paper addresses the identification and evaluation of extreme flood events in the transitional area between Western and Central Europe in the period 1950–2010. Floods are evaluated in terms of several variants on an extremity index that combines discharge values with the spatial extent of flooding. The indices differ in two main aspects: the weight of the

area parameter and the threshold of the considered maximum discharges. This study demonstrates that extremity indices are not highly sensitive to the changes in the design of the area parameter. On the contrary, using the index with a higher flood discharge limit changes the floods' rankings significantly. Due to a positive agreement with high severity events, we recommend using the index with a higher discharge limit.

In general, we detected an increase in the proportion of warm half-year floods when using a higher discharge limit.

Nevertheless, cold half-year floods still predominate in the lists because the affected area is usually large in the case of these floods. This study demonstrates the increasing representation of warm half-year floods from the northwest to the southeast.

## 1 Introduction

Hydrological events, especially floods, are serious natural hazards in Western and Central Europe (Kundzewicz et al., 2005; Munich Re, 2015). Several extreme floods occurred in Western and Central Europe, e.g., in August 2002, January 2003,

March/April 2006, and June 2013. The last was one of the largest in some river basins over the last two centuries (Blöschl et al., 2013).

In addition to river floods, flash floods affect this part of Europe, although these are mostly local events that usually produce less damage (Barredo, 2007). Therefore, we are interested in extensive floods affecting several river basins. Uhlemann et al. (2010) call these floods as trans-basin. They are usually triggered by persistent heavy rainfall and/or snowmelt. Differences in

the causes of river floods can be detected between the western and central parts of Europe. Western Europe experiences flooding primarily during the cold half of the year due to zonal westerly circulation systems (Caspary, 1995; Jacobeit et al., 2003). Towards the east, warm half-year floods become more frequent. This is largely due to cyclones moving along the Vb pathway described by van Bebber (1891). These cyclones move from the Adriatic in a northeastern direction (e.g., Nissen et al., 2014), and the "overturning" moisture flux brings warm and moist air into the central part of Europe (Müller and Kašpar,





2010). However, it is not possible to delineate the borders of Western and Central Europe precisely with respect to differences in their flood events because of a broad transitional zone where both types of flooding occur.

An extremity index is useful for comparing individual flood events and determining their overall extremity. Various indicators and indices are used for the assessment of extreme events (including floods) and in their quantitative comparison. Different

approaches are applied because the definition of event extremity is not uniform (Beniston et al., 2007), so various sets of extreme floods have been compiled in individual papers. The assessment of extreme floods is based on the quantification of human and material losses (severity), high discharge values (intensity), peak discharge return periods (rarity), or a combination of these indicators. The ranking of the largest floods can differ depending on which aspect of extremity was evaluated.

An assessment based on flood severity may be a simple way to evaluate a flood's extremity. Barredo (2007) identified major

flood events in the European Union between 1950 and 2005 to create a catalogue and map of the events. He utilized two simple selection criteria: damage amounting to at least 0.005 % of EU GDP and a number of casualties higher than 70.

Other authors prefer evaluations based on the intensity or rarity of flooding because these aspects better reflect causal natural processes. Some authors classified floods into extremity classes based on the observed water levels (Brázdil et al., 1999; Mudelsee et al., 2003), which is most suitable for long-term pre-instrumental flood records. Water level values for individual

flood events are at our disposal due to high water marks, chronicle records or other documents. This type of flood extremity evaluation was applied to the long-term flood records of the Basel gauge station on the Rhine river (Brázdil et al., 1999) and in the Elbe and Oder river basins (Mudelsee et al., 2003).

Additionally, Rodda (2005) used maximum discharges to express flood extremity in the Czech Republic. He considered the ratio of the maximum mean daily discharge to the median annual flood. This was completed for each station and flood event

to study the spatial correlations among flood intensities in various basins.

Rarity can be used to compare extreme floods at different locations, when extremity is defined not by absolute thresholds (e.g., discharge values) but by relative ones (e.g., n-th quintile of the dataset). Keef et al. (2009) focused on the spatial dependence of extreme rainfall and discharges in the UK and used return periods to define extreme values. Their work confirms that it is possible to compare the event extremities at different locations, even when the actual discharge values vary considerably.

Comprehensive indicators of flood extremity typically combine some aspect of extremity or consider other factors, such as the areal extent or duration of events. When creating these indicators, researchers attempt to add information about flooding from all locations where it was observed. The Francou index k (Francou and Rodier, 1967; Rodier and Roche, 1984; Herschy, 2003) is one of the older indices that assesses flood extremity only at a particular station. In the Francou index, the common logarithm of maximum discharge is divided by the common logarithm of the catchment area (Rodier and Roche, 1984; Herschy, 2003).

Among others, it was used to evaluate the largest floods in the World Catalogue of Maximum Observed Floods (Herschy, 2003).

Müller et al. (2015) designed a more complicated extremity index using return periods of peak discharges. They present 50 maximum floods in the Czech Republic for the period 1961–2010, which are identified based on the so-called flood extremity index (FEI) (Müller et al., 2015). In addition to the peak discharge return periods, the size of the relevant basin is considered



for each location. The authors also suggested extremity indices other than the FEI that are applicable to precipitation events: the weather extremity index (Müller and Kašpar, 2014) and the weather abnormality index. Comparison of these indices with the FEI may aid in examining the relationship between precipitation and flood extremity (Müller et al., 2015).

To analyze the spatial and temporal distribution of floods in Germany, Uhlemann et al. (2010) developed a comprehensive method for the identification and evaluation of major flooding affecting several river basins. They used a time series of mean daily discharges and searched for simultaneously occurring significant discharge peaks comprising individual flood events. Their index accounts for the spatial extent of flooding (expressed by the length of the affected rivers) and discharge peak values exceeding the 2-year return value. The authors present 80 major flood events in Germany from 1952 to 2002.

Subsequently, Schröter et al. (2015) adopted the approach of Uhlemann et al. (2010) and compared several major floods in Germany. Their modified index compiled only those maximum discharges that exceeded the 5-year return value; the discharges were normalized by the respective 5-year return values and weighted by the portion of the affected river length. The final index equals the sum of these values from affected stations. Thus, the indices by Uhlemann et al. (2010) and Schröter et al. (2015) differ only in the threshold of the discharge values entered into the index calculation (2- and 5-year return values, respectively). However, Schröter et al. (2015) presented only the June 2013 flood extremity in comparison with two other major floods in August 2002 and July 1954. Because other major flood events were not presented for comparison, it is not possible to precisely identify the influence of this methodological change on their results.

The main aim of this paper is to determine how changes in the flood evaluation methodology influence the results. The presented indices are based primarily upon the approach of Uhlemann et al. (2010), but their design is somewhat modified. Each of the indices combine the flood discharge magnitude with the spatial extent of flooding; differences lay in the discharge thresholds and input area parameters.

In addition to the sensitivity study, we present lists of extreme flood events from the period 1950–2010 and describe their spatial and temporal distributions. The area of interest might be called "Midwestern" Europe and is basically a transitional area between Western and Central Europe. It has natural boundaries: the Alps to the south, the Carpathian and Sudeten Mountains to the east and the coast of the North Sea to the northwest. The area is defined by six main river basins: Rhine, Elbe, Meuse, Weser, Ems, and Danube up to Bratislava. As mentioned above, this area is interesting because of a noticeable shift in the seasonality of floods in a west to east direction. Due to its heterogeneity and vastness, the area is also convenient for index design assessment when evaluating the extremity of floods affecting several river basins.

## 2 Data and methods

### 2.1 Data

We used mean daily discharge values at selected stations (for each day during the period 1950–2010) as a basis when searching for floods that occurred simultaneously within the study area. Only data from stations enclosing at least 2500 km$^2$ of the relevant river basin were used due to poor data availability for smaller catchments and to exclude minor floods. This work is



based primarily on data that were obtained from the database of the Global Runoff Data Centre (GRDC), an international archive of monthly and daily discharges. The time series was incomplete in some cases, so we used additional data from national hydrological yearbooks and the Czech Hydrometeorological Institute. When necessary, missing values were obtained using linear regression; only one or two missing years were supplied in this manner.

As a result, 115 gauging stations from six countries (the Czech Republic, Slovakia, Austria, Switzerland, Germany and the Netherlands) were selected to analyze the time series of mean daily discharges between 1950 and 2010. The study area is 499149 km$^2$, the average size of a catchment is 31293 km$^2$, with the Lobith station enclosing the maximum catchment at 160800 km$^2$. Only the lower part of a catchment was related to a given station when another station was located upstream. The selected stations and their subcatchments are depicted in Fig. 1. The size of the subcatchments ranges from 248 to 21301 km$^2$,

with a mean area of 4340 km$^2$. The spatial distribution of gauging stations in the dataset is not entirely uniform: the density of stations is highest in the Weser river basin, and the Meuse river basin has the least coverage.

## 2.2 Methods

### 2.2.1 Identification of significant mean daily discharges

The first step in this study is the selection of flood peaks at individual stations. The local maxima within the time series of

mean daily discharges ($Q_d$) must be identified. Local maxima are $Q_d$ values that are higher than values on both the previous and the following day. If several consecutive days have exactly the same value of $Q_d$, the first day is used.

For each gauging station, most sets of local maxima are due to minor flow fluctuations. To select significant discharges ($Q_s$), the local maxima are compared with the mean annual maximum of mean daily discharges ($Q_{ma}$) calculated from $n = 61$ annual maxima of mean daily discharges ($Q_a$) at a station $i$:

$$Q_{mai} = \frac{1}{n} \sum_{j=1}^{n} Q_{aij}.$$
(1)

Discharges exceeding $Q_{ma}$ are considered significant. Nevertheless, we assume that a serious flood must be characterized by even higher discharges at least in a part of the affected area. Therefore, we also search for peak discharges that are equal to or greater than the 10-year return level of mean daily discharge ($Q_{10}$). The values of $Q_{10}$ were estimated from the series of $Q_a$ values that were approximated by the generalized extreme value distribution (GEV) using the L-moments method.

### 2.2.2 Determination of significant flood events

A significant flood event is defined here as a group of time-related $Q_s$ at various stations where at least one $Q_s$ value equals or exceeds $Q_{10}$. However, peak discharges often do not occur exactly on the same day due to, e.g., the extent of the study area, the propagation of flood waves downstream, or the movement of the precipitation field. Therefore, a time window when $Q_s$ values seem to belong to the same event is defined. After analyzing all of the data series, we chose a time window that includes





ten days before and ten days after the occurrence of the first value of $Q_d \geq Q_{10}$. If there are other values of $Q_d \geq Q_{10}$ within that time span, the time window is further extended with respect to the date of this peak discharge. This time span is slightly longer than that used by Uhlemann et al. (2010), but this difference is reasonable because a larger area is studied here. Moreover, the values of $Q_s$ systematically lag behind at hydrometric profiles on the Havel River and its largest tributary the Spree. This may

be due to the lowland character of these basins permitting extensive spilling of water. We therefore decided to extend the time delay at these stations up to 12 days. However, the chosen time window may be too long in some cases because another atmospherically unrelated event may begin.

Therefore, we introduce an additional rule for dividing flood peaks that were identified as time-related but are in fact associated with different atmospheric causes. If more $Q_s$ values are identified in a time series within an individual flood event and the

time span between those peaks is at least five days long, we divide the peaks into two floods; otherwise, only one flood event is considered. Finally, only the highest $Q_s$ in a time series is considered.

### 2.2.3 Extremity indices design

Over 200 significant flood events are identified in the period 1950–2010. First, they are evaluated only with respect to the size of the affected area:

$$A = \sum_{i=1}^{k} a_i \tag{2}$$

where $a_i$ denotes the area of one of $k$ subcatchments where $Q_s$ is detected. The 80 largest floods are further examined. First, they are sorted based on whether they occurred during the colder or the warmer half of the year; the decisive day for classification is the first day with $Q_s$. The colder half-year is set between 16 October and 15 April because there is evidence from the Czech Republic of a relatively sharp interface in terms of flood occurrence in mid-April (Müller et al., 2015).

Both the spatial extent of floods and the aspect of the discharge magnitudes must be incorporated into an extremity index for evaluating extreme flood events. To demonstrate the role of the weights of both aspects, we defined nine index variants with differences in input parameters and applied them to the 80 selected floods.

Generally, the index is derived from $A$ by multiplying $a_i$ (or its function) by normalized peak discharges. Three basic variants consider all of the $Q_s$ values normalized by the respective value of $Q_{ma}$, but they vary in the area parameter. The first variant

simply considers subcatchment areas $a_i$:

$$E_a = \sum_{i=1}^{k} \left( \frac{Q_{si}}{Q_{mai}} a_i \right) \tag{3}$$

whereas two other variants contain the square root of each subcatchment area and their common logarithm, respectively:





$$E_{\mathrm{r}} = \sum_{i=1}^{k} \left( \frac{Q_{\mathrm{si}}}{Q_{\mathrm{mai}}} \sqrt{a_{\mathrm{i}}} \right),$$  (4)

$$E_{\mathrm{l}} = \sum_{i=1}^{k} \left( \frac{Q_{\mathrm{si}}}{Q_{\mathrm{mai}}} \log a_{\mathrm{i}} \right).$$  (5)

Using the square root in Eq. (4) reduces the weight of the aspect of the affected area. When applying the logarithm of the area in Eq. (5), the reduction is even more significant because the range of possible parameter values is markedly reduced. As a result, the role of discharge magnitudes in the index increases in Eq. (4) and even more so in Eq. (5).

Another way to modify the extremity index is to set a different threshold of considered discharge values. Although all $Q_{\mathrm{s}}$ values are used in the three basic variant calculations, three other variants labeled $E_{1.2\mathrm{a}}$, $E_{1.2\mathrm{r}}$, and $E_{1.2\mathrm{l}}$ consider discharges that fulfill the condition $Q_{\mathrm{s}}/Q_{\mathrm{ma}} > 1.2$; they are determined by Eqs. (3), (4), and (5). The remaining three variants labeled $E_{1.5\mathrm{a}}$, $E_{1.5\mathrm{r}}$, and $E_{1.5\mathrm{l}}$ are analogous but the threshold is augmented up to 1.5.

These indices suppress the influence of the size of the flood area in the final extremity index value, and the discharge values of a flood event become more emphasized. When calculating indices with higher discharge thresholds, the total number of flood events may be reduced due to removing some flood events with rather low mean daily discharges.

Finally, we select 30 major floods according to each of the nine extremity index variants. As the total study period covers 61 years, we select one flood per two years on average. This enables a comparison of the rankings of flood events with respect to the individual index variants. This comparison opens the discussion of the role of extremity index design.

## 3 Results

The 80 largest floods affected an area between 81000 and 381000 km², which is between 16 % and 76 % of the area of interest (Fig. 2). The dominance of the cold half-year floods is obvious, especially in the first half of the chart in Fig. 2. The largest flood occurred at the turn of March and April 1988. It is clear that the warm half-year floods relate more to the Danube basin or eventually to the Elbe river basin. The Rhine river basin is less represented and in the Meuse, Weser and Ems river basins, warm half-year floods rarely occur.

### 3.1 Comparison of the extremity index variants

Figure 3 depicts differences among the extremity index variants in terms of their dependence on the size of the affected area $A$. Each chart in Fig. 3 represents one variant of the extremity index. The correlation between $A$ and the index values is much higher when the threshold $Q_{\mathrm{s}}/Q_{\mathrm{ma}}$ is set to 1. Surprisingly, this close correlation persists, even when the common logarithm of the catchment areas is applied (Fig. 3a at the right). There are similarities among rankings of the events with respect to both $A$ and the index values. The only exception is the August 2002 flood with a relatively small value of $A$. The rankings of the three




highlighted flood events remain close, regardless of the variant. This is also shown in Fig. 3b representing index variants with the threshold $Q_s/Q_{ma} = 1.2$. Nevertheless, the correlation between $A$ and the index values is lower, which is even more obvious when the threshold $Q_s/Q_{ma} = 1.5$. Still, we can see only minor differences among the indices with the same discharge threshold but a different area parameter shape.

In summary, the dependence of flood extremity on the size of the affected area does not change significantly with changes in the area parameter. This indicates that the index is not highly sensitive to changes in the area parameter but is instead related to the discharge threshold. If the threshold of $Q_s/Q_{ma}$ rises to 1.2 or even 1.5, only stations with greater flood discharges are included in the calculation. The influence of the affected area of flooding is suppressed in these indices, and the flood extremity should relate in particular to the flood discharges reached.

## 3.2 Major floods

Two variants of the index ($E_r$ a $E_{1.5r}$) were chosen to create the final lists of 30 major floods in the transitional area between Western and Central Europe in the period 1950–2010 (Tables 1 and 2, respectively). They both employ the middle variant of the area parameter, i.e., the square root of $A$, which makes them similar to the variants using either the actual area or its logarithm. Nevertheless, the chosen variants differ significantly in the discharge threshold.

Floods selected by $E_r$ are primarily extensive events as small flood discharges are also considered. The flood of March/April 1988 is the first of the major floods, followed by the January flood of 2003 and the flood of August 2002. Overall, there are only four events in the warm half-year among 30 maxima. On the contrary, the list of floods according to the $E_{1.5r}$ is more balanced from this point of view. It contains seven events that are not present among the maxima according to $E_r$; five of these extra floods belong to the warm half-year. These floods replaced some cold half-year floods with relatively low values of $Q_d$.

More floods with lesser extents are present in the list in Table 2. Nevertheless, four floods were evaluated as being at the maximum, regardless of the index variant, with only different ranking among them; the August flood of 2002 is the biggest according to the $E_{1.5r}$ due to its extremely high discharge values.

### 3.2.1 Seasonal distribution

Floods of the cold half-year are generally better represented among the major flood events. The seasonal distribution is quite
similar for $E_r$ and $E_{1.5r}$, with a frequency maximum in winter and a secondary maximum in summer (Fig. 4). According to $E_r$, major events are concentrated from January to March, but they are spread more equally from December to April according to $E_{1.5r}$. This indicates that the first half of April is characterized by floods with rather small spatial extents. The secondary frequency maximum occurs in July and August and is much more pronounced according to $E_{1.5r}$. The rest of the year is characterized by a low frequency of major floods. Although one event per calendar month was recorded in both May and June,
only a single major flood occurred from late August to the beginning of December. It began at the end of October 1998, and its extremeness was surprisingly high, according to both variants of the extremity index.





### 3.2.2 Interannual variability

Major floods do not occur regularly over time. Some clusters of flood events are apparent in Fig. 5, which presents the distribution of major floods between 1950 and 2010. The July flood of 1954 is the first recorded flood in the period examined. A significant accumulation of flooding is apparent in the 1980s and from 1993 to 2006. On the contrary, a long period without

major floods occurred at the beginning of the 1960s.

Generally, there are more major floods in the second half of the period, which applies to both index variants. It seems that the number of events is increasing, as is their extremity. However, the extremity according to $E_{1.5r}$ is increasing more rapidly, which may be due to a higher number of warm half-year floods towards the end of the study period.

### 3.2.3 Spatial distribution

Regarding the spatial distribution of floods, Fig. 2 demonstrates that floods during the warm half-year relate more to the Danube and the Elbe river basins. Warm half-year floods are less frequent in the Rhine river basin, and they occur very rarely in the Meuse, Weser and Ems river basins, where cold half-year floods dominate. This is confirmed by Fig. 6, which depicts the frequency of 30 major floods in both half-years within individual subcatchments.

In general, the number of cold half-year floods decreases towards the southeast, whereas the number of warm half-year floods

increases in the same direction. Regardless the variant of the extremity index, there are regions affected by extreme floods only in one part of the year. This is true for the Meuse, Weser, Ems, and the lower part of the Rhine river basin including Main (cold half-year) and most of the Alpine rivers (warm half-year). On the contrary, other regions are prone to extreme floods both in the cold and the warm halves of the year: the Elbe and Danube river basins, apart from the Alpine tributaries. Differences among the variants of the index exist only in the numbers of flood events in individual subcatchments.

**4 Discussion and conclusions**

This paper addresses the evaluation of major flood events in the transitional area between Western and Central Europe in the period 1950–2010. Major floods are defined according to the value of a flood extremity index. We created nine variants of the index with differences in terms of design, specifically regarding discharge thresholds and area parameters. We were motivated by Uhlemann et al. (2010) and Schröter et al. (2015), who used similar flood extremity indices, with only a difference in the

threshold of the discharge values entered into the calculation. Uhlemann et al. (2010) used a 2-year flow threshold, which corresponds approximately to the value of $Q_{ma}$, or is slightly lower. Schröter et al. (2015) chose a higher limit of a 5-year flow, thus making these studies incomparable. In this paper, we introduce the differences that arise in the resulting lists of major floods when we use indices with different discharge thresholds and area parameters. We selected the value of $Q_{ma}$ as a basic threshold and two additional threshold values designed as multiples of $Q_{ma}$. We found that the value of this threshold is crucial

for the ranking of major floods. The number of warm half-year floods increases in the lists of major floods when using the higher discharge thresholds. On the contrary, the index variants are not highly sensitive to changes in the area parameter. Two



sets of 30 major floods are presented according to their $E_r$ and $E_{1.5r}$ indices, and the respective lists are compared in terms of seasonality, interannual variability and spatial distribution.

Regarding the seasonal distribution of major flood events, the predominance of cold half-year floods is apparent in both lists. Uhlemann et al. (2010) showed the same result. In contrast, floods during the warm half of the year dominate the list of the 30 major floods in the Czech Republic by Müller et al. (2015). This may be due to the fact that the occurrence of warm half-year floods is increasing from the northwest to the southeast in the studied area. The list of major floods for the Czech Republic is closer to the list based on $E_{1.5r}$ because warm half-year floods are better covered by the index variant that has the higher discharge threshold.

The temporal distribution of major flood events during the period between 1950 and 2010 is rather uneven. There are certain clusters in terms of the occurrence of major floods. Some periods of reduced or increased frequencies of major flooding are identical to the results of other papers (Uhlemann et al., 2010; Müller et al., 2015). For example, we found these identical trends: a higher frequency of major floods in the 1980s and a decline in the number of identified floods in the 1990s. The last five-year period between 2006–2010 is different, however, because it is a period with a higher frequency of major flooding in Müller et al. (2015). The increase in major flooding in the second half of the period is again consistent with the findings of Uhlemann et al. (2010). However, it remains unclear whether this is a trend or just a part of a cycle.

Generally, the lists of major floods are quite similar to the list of German trans-basin floods presented by Uhlemann et al. (2010) because Germany covers more than half of the area studied in this work. The consensus is greater in the case of the $E_r$ index. The duration of "identical" floods is slightly different, as is their ranking. This is due to the different size of the area of interest and the flood identification methodology. Schröter et al. (2015) used an index similar to Uhlemann et al. (2010), but the authors only offered a comparison of the extremity of three summer flood events: the floods of 1954, 2002 and 2013. The flood event of 2013 is reported as the largest, followed by the flood of 1954. In this paper, the flood of August 2002 is always more extreme than the flood of 1954, regardless of the index variant used, because of the differences in the extent of the area of interest.

We can also compare our results with those of Barredo (2007), who provided a set of 21 large European river floods compiled according to the amount of damage caused. Six of these floods affected our area of interest; all are included in the set of major floods according to $E_{1.5r}$, but only three belong to the 30 major events with respect to $E_r$. Obviously, floods that caused major damage are better represented by the variant of the extremity index with a higher threshold of considered discharge values. Therefore, we recommend using $E_{1.5r}$ for the evaluation of extensive floods. The $E_{1.5r}$ index is apparently better able to identify major floods.

Further research on this topic will examine the related atmospheric conditions. A comprehensive evaluation of causal circulation conditions, the consequent precipitation and the flow response is needed. A comparison of major floods with precipitation and circulation extremes would be useful for a better understanding of the causes of extensive floods which affect several river basins.





*Acknowledgements.* The presented research is supported by the Charles University (project GA UK 276816). We thank the Global Runoff Data Centre and the Czech Hydrometeorological Institute for providing runoff data.

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





**Table 1: List of 30 major floods according to the $E_r$ index. The date is displayed in the YYYY/MM/DD format. The A/S column refers to the proportion of the affected catchment size to the total size of the area of interest. Warm half-year floods are in bold.**

| Ranking | Flood duration | | | $E_r$ | A/S [%] |
|:---:|:---|:---:|:---|:---:|:---:|
| 1 | 1988/03/25 | - | 1988/04/08 | 7948 | 76 |
| 2 | 2003/01/03 | - | 2003/01/23 | 7299 | 65 |
| **3** | **2002/08/11** | **-** | **2002/08/23** | **6621** | **44** |
| 4 | 1981/03/11 | - | 1981/03/30 | 6351 | 54 |
| 5 | 1956/03/03 | - | 1956/03/20 | 5961 | 65 |
| 6 | 1982/01/06 | - | 1982/01/17 | 5930 | 63 |
| 7 | 1970/02/23 | - | 1970/02/28 | 5708 | 57 |
| 8 | 1995/01/25 | - | 1995/02/12 | 5661 | 55 |
| 9 | 1998/10/29 | - | 1998/11/13 | 5423 | 55 |
| 10 | 2006/03/29 | - | 2006/04/09 | 5289 | 43 |
| 11 | 1993/12/21 | - | 1994/01/03 | 5165 | 51 |
| 12 | 1987/01/01 | - | 1987/01/10 | 5053 | 49 |
| 13 | 1980/02/05 | - | 1980/02/24 | 4761 | 56 |
| **14** | **1954/07/09** | **-** | **1954/07/21** | **4569** | **40** |
| 15 | 2002/02/27 | - | 2002/03/15 | 4239 | 49 |
| **16** | **1965/06/10** | **-** | **1965/06/20** | **4207** | **44** |
| 17 | 1988/03/17 | - | 1988/03/26 | 4125 | 46 |
| 18 | 1994/04/13 | - | 1994/04/23 | 3959 | 38 |
| 19 | 1968/01/16 | - | 1968/01/27 | 3817 | 41 |
| 20 | 1987/03/26 | - | 1987/04/06 | 3392 | 33 |
| 21 | 1974/12/08 | - | 1974/12/18 | 3253 | 34 |
| 22 | 1999/03/03 | - | 1999/03/13 | 3212 | 37 |
| **23** | **1981/07/20** | **-** | **1981/07/29** | **3105** | **31** |
| 24 | 1982/01/31 | - | 1982/02/05 | 2866 | 36 |
| 25 | 1994/01/02 | - | 1994/01/16 | 2843 | 31 |
| 26 | 1967/12/24 | - | 1967/12/28 | 2802 | 31 |
| 27 | 1984/02/07 | - | 1984/02/12 | 2757 | 30 |
| 28 | 2002/03/21 | - | 2002/03/25 | 2754 | 35 |
| 29 | 1979/03/12 | - | 1979/03/30 | 2725 | 32 |
| 30 | 1955/01/14 | - | 1955/01/25 | 2621 | 30 |



**Table 2: Same as Table 1 but for the $E_{1.5r}$ index.**

| Ranking | Flood duration | | | $E_{1.5r}$ | A/S [%] |
|---|---|---|---|---|---|
| **1** | **2002/08/11** | - | **2002/08/23** | **3580** | **44** |
| 2 | 2003/01/03 | - | 2003/01/23 | 2926 | 65 |
| 3 | 1981/03/11 | - | 1981/03/30 | 2442 | 54 |
| 4 | 1988/03/25 | - | 1988/04/08 | 2256 | 76 |
| 5 | 1995/01/25 | - | 1995/02/12 | 2150 | 55 |
| **6** | **1954/07/09** | - | **1954/07/21** | **2026** | **40** |
| 7 | 2006/03/29 | - | 2006/04/09 | 2021 | 43 |
| 8 | 1993/12/21 | - | 1994/01/03 | 1720 | 51 |
| 9 | 1970/02/23 | - | 1970/02/28 | 1621 | 57 |
| 10 | 1987/01/01 | - | 1987/01/10 | 1351 | 49 |
| 11 | 1998/10/29 | - | 1998/11/13 | 1166 | 55 |
| **12** | **1965/06/10** | - | **1965/06/20** | **1092** | **44** |
| **13** | **1981/07/20** | - | **1981/07/29** | **973** | **31** |
| **14** | **1999/05/16** | - | **1999/05/26** | **936** | **23** |
| 15 | 1982/01/06 | - | 1982/01/17 | 892 | 63 |
| 16 | 1956/03/03 | - | 1956/03/20 | 819 | 65 |
| **17** | **2005/08/22** | - | **2005/08/26** | **811** | **22** |
| 18 | 1994/04/13 | - | 1994/04/23 | 782 | 38 |
| 19 | 1988/04/01 | - | 1988/04/14 | 608 | 21 |
| **20** | **1958/07/05** | - | **1958/07/16** | **576** | **21** |
| 21 | 1988/03/17 | - | 1988/03/26 | 575 | 46 |
| 22 | 1974/12/08 | - | 1974/12/18 | 566 | 34 |
| 23 | 1955/01/14 | - | 1955/01/25 | 504 | 30 |
| 24 | 1984/02/07 | - | 1984/02/12 | 499 | 30 |
| 25 | 1983/04/09 | - | 1983/04/16 | 483 | 24 |
| 26 | 1968/01/16 | - | 1968/01/27 | 430 | 41 |
| **27** | **1997/07/06** | - | **1997/07/10** | **428** | **21** |
| **28** | **1985/08/06** | - | **1985/08/12** | **427** | **19** |
| 29 | 1967/12/24 | - | 1967/12/28 | 419 | 31 |
| 30 | 1987/03/26 | - | 1987/04/06 | 382 | 33 |



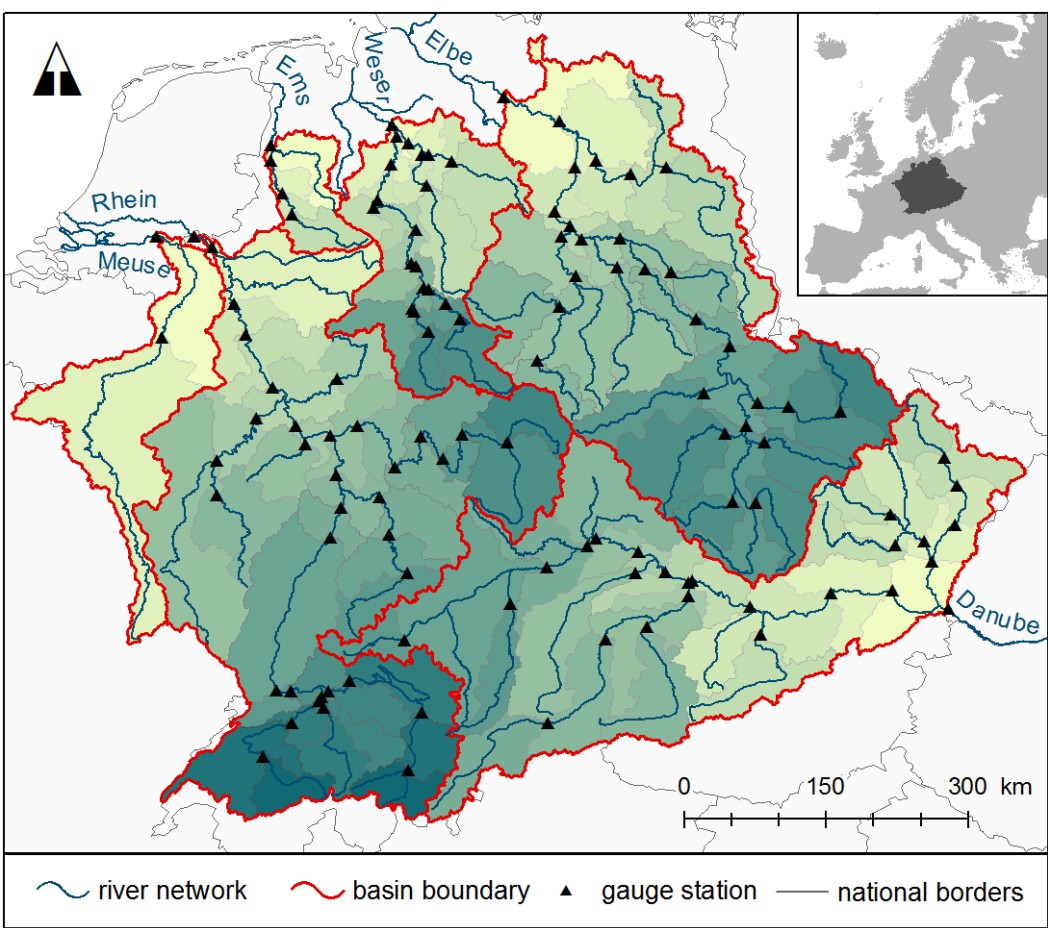

**Figure 1: Gauge stations in the area of interest. Their subcatchments are distinguished by color.**



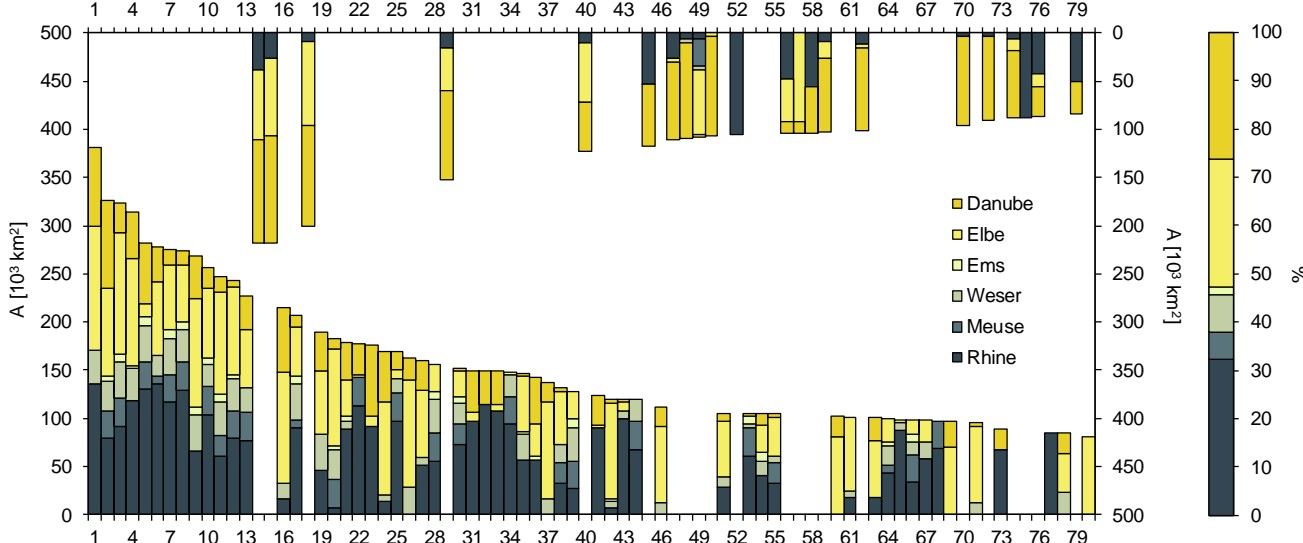

**Figure 2: The 80 largest flood events in the study area from 1950 to 2010 according to the size of the affected area $A$. Cold and warm half-year floods are depicted on the bottom and the top x-axis, respectively. Contributions of individual river basins to the flood event area are distinguished by color. Their contribution to the total area of interest is shown in the right bar.**



**Figure 3: Relationship between the affected area A (x-axis) and the flood extremity according to nine index variants. Solid lines depict linear trends in the data; R² indicates the value of the coefficient of determination. Selected floods are highlighted: March/April 1988 (1); January 2003 (2); August, 2002 (3).**

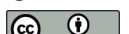



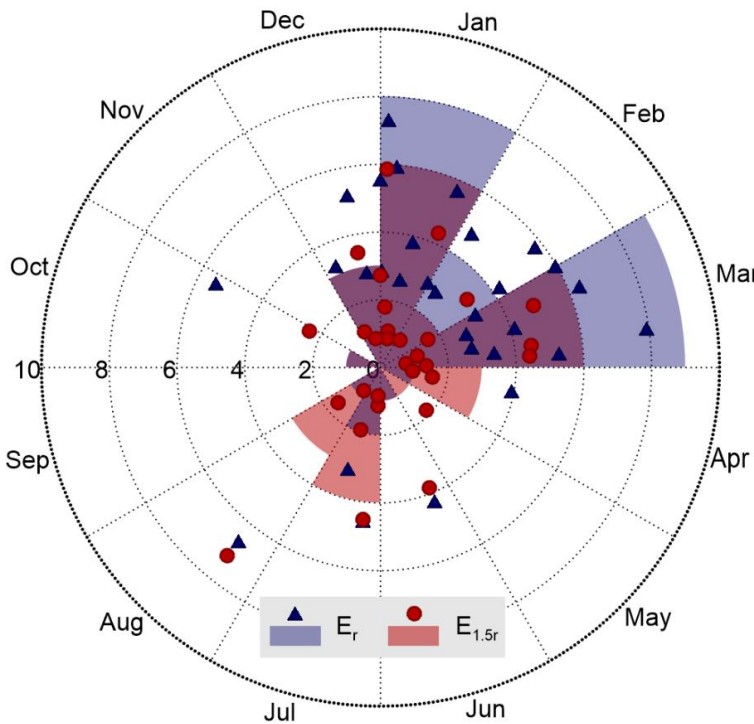

**Figure 4: Seasonal distribution of 30 maximum floods according to $E_r$ and $E_{1.5r}$ indices. The number of extreme floods in individual months is depicted by shading; the mixed color indicates overlapping data. The signs represent calendar days when individual floods began; the distance of the sign from the center of the diagram reflects the flood extremity given by the value of $E_r$ [$10^3$] and $E_{1.5r}$ [$5 \cdot 10^2$].**




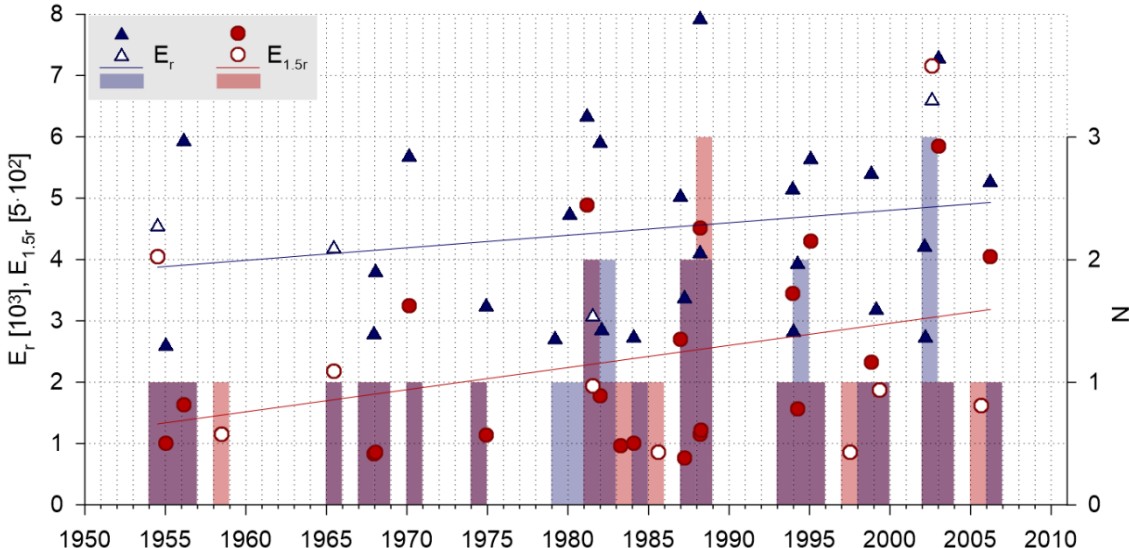

**Figure 5: Interannual variability of 30 major floods according to $E_r$ and $E_{1.5r}$. The number $N$ of major floods in individual years is depicted by shading; the mixed color indicates overlapping data. The symbols represent the extremity of cold half-year floods (solid symbols) and warm half-year floods (hollow symbols) with respect to $E_r$ [$10^3$] and $E_{1.5r}$ [$5 \cdot 10^2$]; Solid lines depict the linear trend in the flood extremity.**





**Figure 6: Spatial distribution of 30 maximum floods according to $E_r$ (a, b) and $E_{1.5r}$ (c, d). The numbers of cold (a, c) and warm (b, d) half-year floods in individual subcatchments during 1950–2010 are depicted by shading.**