# Peer review of "Evaluation of extensive floods in Western/Central Europe"

_Hydrology and Earth System Sciences, 2016_

## Referee Comment (RC1) · Anonymous Referee #1 · 15 Dec 2016

The paper addresses the topic of identification of flood events in western and central Europe based on multiple series of mean daily discharge for a period of 61 years. Building on the method of Uhlemann et al. (2010) the authors aim to highlight the influence of different parameter choices on the the flood severity index which in turn influences the event identification.

Extending the analysis of Uhlemann et al. 2010 to a larger study area, i.e. identifying a true central European event set of large flood events is a valid objective. However, the paper lacks scientifc rigour and presents little novelty on the event severity assessment. I therefore do not recommend the paper for further publication in HESS. In the following, I will outline my key criticism and encourage the authors to commence with their research on the important topic of understanding flood event frequency, severity and causes in central Europe.

Uhlemann present a thorough sensitivity analysis of the severity index already and present the impact of different thresholds and input data on the resulting event set. The work presented by Gvoždíková only addresses the sensitivity of two parameters: subcatchment area and flood discharge limit (which is a threshold of flow expressed as the ratio of the peak flow at a gauge against the mean annual maximum flow of the entire time series at that gauge). The selection of the parameters and the chosen range in which they are being tested is not supported by argument. I.e., what is the hypothesis for defining the three variants of the thresholds of what is called the discharge limit ($Q_s/Q_{ma}$; $Q_s/Q_{ma}>1.2$; $Q_s/Q_{ma} >1.5$)? Likewise, choosing either the subcatchment area, its root or logarithm as range for testing the impact of the spatial weight on the severity index is arbitrary. The event sets identified are limited in a first step to 80 events, and then, for comparing results of the different variants of the index, to 30. Why is that so? My strongest criticism is with the principle choice of subcatchment area as spatial weighting factor to account for the relative contribution of a peak recorded at a river gauge to the overall flood severity. This is a classical regionalisation problem in hydrology. Subcatchment area however fails to address this problem and introduces a severe spatial bias into the analysis. Unlike precipitation, for which area indices are well suited, floods are not a space-filling phenomena. In particular, peak flows recorded at downstream gauges of the large streams Rhine, Elbe, to some degree Danube, Weser, and Meuse are in most cases not caused by inflow from their intermittent catchments but are a result of the flood wave propagating from further upstream. Also, when choosing subcatchment area, the density of the gauge network and particularly the uneven distribution of gauges in the river network becomes relatively more important in the severity index calculation and needs sensitivity testing. The original severity index presented by Uhlemann 2010, and also the application of the index in Schröter et al. 2015, provide a method for regionalisation of peak flows to the river network rather than to the subcatchments. I strongly recommend the authors to review their method for computing the spatial extent of the flood events in any further study. In summary, the conclusions reached on the best suited variant of the severity index and

resulting event set need thorough reworking. In fact, I think, that the sensitivity study provided in Uhlemann et al 2010 provides all the necessary findings to allow for a fairly straight forward adoption of the severity index to the context of identifying flood events in central Europe. In the final paragraph of the paper the authors highlight that they want to commence with an analysis of the hydro-meteorological causes of large flood events in central Europe. I think, this is where the innovation will come and I highly encourage the authors to proceed on this avenue. The assessment of the severity of events and consequently the identification of the relevant flood events in the region can be natural part of any paper submitted on this.

On the aspect of analyzing severe floods in W/Central Europe in their frequency and severity and also in their spatial-temporal patterns and potential changes of these + attribution of these changes to causes: Reading the title I had expected to see the Odra basin included in the study. This basin forms the eastern boundary of the very wide transitional zone between atlantic and continental influences on flood genesis and at present I expect that in particular some of the summer flood events are insufficiently represented in the event set(s). Also, extending the event analysis to the most recent period, e.g. 2015, would add value to any change detection and finally attribution. In principle, I think Central Europe is the better description for the area under study.

---

## Referee Comment (RC2) · Anonymous Referee #2 · 5 Jan 2017

The study presented in this paper aims at redefining flood extremity indices over a large region between Western and Central Europe over the period 1950–2010. The approach followed consists of designing flood extremity indices by combining discharge values and the spatial extent of floods. Several versions of such indices were tested, with different weightings of the threshold value of discharge or area parameter for considering a flood event. The topic is suitable for publication in HESS but major revisions would be necessary in my opinion before the paper be published.

General: - The paper lacks a discussion on the consistency of the choices to be made for designing the extremity indices (determination of Qs and the threshold Qs/Qma). There is almost no discussion about this point which constitutes the basis of the whole approach. Also, it seems from the discussion/conclusion section that the main difference between this work and previous other ones upon which the present study builds

relies on the choice of the threshold selected for discharge: 1) this emphasizes even more the importance of strengthening the discussion on criteria for choosing the best suited Qs/Qma threshold, 2) it questions the value-added of including an area parameter in the approach (the authors themselves state that extremity indices are not very sensitive in changes in the area parameter: if so, then this approach is very similar to previous ones?).

- Conducting a more detailed study on the determinism of the occurrence of flood events seems important in order to relate the extremity indices defined to more concrete or practical hydrological/hydrometerological processes (in this sense it is surprising that the role of ground water is never even mentioned), it should be addressed here and would certainly constitute the value-added to other previous studies such as those of Uhlemann et al., etc. As a first step, the authors could try to relate the interannual variability and trends of extremity indices to some climate indices for instance.

Specific comments: - Title: Something like "large spatial extent floods" or "extensive floods" (as used in the introduction) could be included in the title to be more specific as it is an important aspect, and would prevent from using "trans-basin" which indeed could be misleading?

- P.4, line 8: I don't get why only the downstream sub-catchment area is considered when an upstream gauging station is available. The downstream station is still representative of flow occurring over the whole upstream area anyway unless the upstream part of flow is substracted.

P.4-5, "Methods": I do not recommend using the word "significant" in this context, as this does not refer to any statistical meaning here. I think the rationale for using this method to determine a time series of "significant" discharge values lacks explanation. As well, I am concerned by the approach for determining a significant flood event: the choice of the length of the time window needs a little more explanation. As is, it looks like the method suffers from a lack of either statistical or deterministic basis, and the

definition of a flood event seems to be too much data- and operator-dependent and not enough transposable (see for instance "After analyzing all of the data series, we chose a time window [...]"). The fact that the time window had to be extended for one river, or that an additional rule had to be included to prevent merging events that have different atmospheric origins is also problematic: is an automatic split of flood events in two parts when they are separated by 5 days enough to conclude to different atmospheric causes ?

P.5, line 19: does the separation date between the cold and warm halves of the year also hold for other regions than the Czech Republic?

---

## Author Response (AR1)

**Responses to reviewer's comments**

First, we want to thank both reviewers for their comments and suggestions. On their basis, we have made several changes in the manuscript. Generally, we adjust our methodology to the method of Uhlemann et al. (2010) and Schröter et al. (2015) for our study area, compare different discharge limits and present a set of extreme floods with their main spatial and temporal characteristics. We add Oder basin into the evaluation of floods and adapt time period to 1951−2013.

**Referee 1 comments:**

**Referee:** The paper addresses the topic of identification of flood events in western and central Europe based on multiple series of mean daily discharge for a period of 61 years. Building on the method of Uhlemann et al. (2010) the authors aim to highlight the influence of different parameter choices on the the flood severity index which in turn influences the event identification. Extending the analysis of Uhlemann et al. 2010 to a larger study area, i.e. identifying a true central European event set of large flood events is a valid objective. However, the paper lacks scientifc rigour and presents little novelty on the event severity assessment. I therefore do not recommend the paper for further publication in HESS. In the following, I will outline my key criticism and encourage the authors to commence with their research on the important topic of understanding flood event frequency, severity and causes in central Europe. Uhlemann present a thorough sensitivity analysis of the severity index already and present the impact of different thresholds and input data on the resulting event set. The work presented by Gvoždíková only addresses the sensitivity of two parameters: subcatchment area and flood discharge limit (which is a threshold of flow expressed as the ratio of the peak flow at a gauge against the mean annual maximum flow of the entire time series at that gauge). The selection of the parameters and the chosen range in which they are being tested is not supported by argument. I.e., what is the hypothesis for defining the three variants of the thresholds of what is called the discharge limit ($Qs/Qma$; $Qs/Qma > 1.2$; $Qs/Qma > 1.5$)? Likewise, choosing either the subcatchment area, its root or logarithm as range for testing the impact of the spatial weight on the severity index is arbitrary.

**Authors:** Thank you for your comment that our main objective – evaluation of flood extremes without limits of state borders – is valid. Our motivation for modifications of the Uhlemann et al. (2010) index was the future comparison with precipitation extremes where the affected area is a necessary parameter. Nevertheless, we accept your comments regarding the methodology, see our next response. The threshold is set to return levels in the reconstructed paper.

**Referee:** The event sets identified are limited in a first step to 80 events, and then, for comparing results of the different variants of the index, to 30. Why is that so?

**Authors:** The first selection was done in order to eliminate less extensive floods. This step is removed. The set of 30 extreme floods is finally presented as we wanted to select one flood per two years on average.

**Referee:** My strongest criticism is with the principle choice of subcatchment area as spatial weighting factor to account for the relative contribution of a peak recorded at a river gauge to the overall flood severity. This is a classical regionalisation problem in hydrology. Subcatchment area however fails to address this problém and

introduces a severe spatial bias into the analysis. Unlike precipitation, for which area indices are well suited, floods are not a space-filling phenomena. In particular, peak flows recorded at downstream gauges of the large streams Rhine, Elbe, to some degree Danube, Weser, and Meuse are in most cases not caused by inflow from their intermittent catchments but are a result of the flood wave propagating from further upstream. Also, when choosing subcatchment area, the density of the gauge network and particularly the uneven distribution of gauges in the river network becomes relatively more important in the severity index calculation and needs sensitivity testing. The original severity index presented by Uhlemann 2010, and also the application of the index in Schröter et al. 2015, provide a method for regionalisation of peak flows to the river network rather than to the subcatchments. I strongly recommend the authors to review their method for computing the spatial extent of the flood events in any further study.

**Authors:** We fully accept your comment and we recalculated our results with respect to the methodology by Uhlemann et al. (2010).

**Referee:** In summary, the conclusions reached on the best suited variant of the severity index and resulting event set need thorough reworking. In fact, I think, that the sensitivity study provided in Uhlemann et al 2010 provides all the necessary findings to allow for a fairly straight forward adoption of the severity index to the context of identifying flood events in central Europe. In the final paragraph of the paper the authors highlight that they want to commence with an analysis of the hydro-meteorological causes of large flood events in central Europe. I think, this is where the innovation will come and I highly encourage the authors to proceed on this avenue. The assessment of the severity of events and consequently the identification of the relevant flood events in the region can be natural part of any paper submitted on this.

**Authors:** Yes, analysis of hydrometeorological causes of central European floods will be the next step of our research. The focus of the reconstructed paper shift from the methodological issue to the analysis of the set of flood events. However, the discussion of the role of threshold remain a part of the paper because the thresholds play an important role in evaluation of floods within the large central European region.

**Referee:** On the aspect of analyzing severe floods in W/Central Europe in their frequency and severity and also in their spatial-temporal patterns and potential changes of these + attribution of these changes to causes: Reading the title I had expected to see the Odra basin included in the study. This basin forms the eastern boundary of the very wide transitional zone between atlantic and continental influences on flood genesis and at present I expect that in particular some of the summer flood events are insufficiently represented in the event set(s). Also, extending the event analysis to the most recent period, e.g. 2015, would add value to any change detection and finally attribution. In principle, I think Central Europe is the better description for the area under study.

**Authors:** We agree that Odra river should be studied together with other central European rivers. We extended the study area and partly also the study period according to your comment.

**Referee 2 comments:**

**Referee:** The study presented in this paper aims at redefining flood extremity indices over a large region between Western and Central Europe over the period 1950–2010. The approach followed consists of designing flood extremity indices by combining discharge values and the spatial extent of floods. Several versions of such indices were tested, with different weightings of the threshold value of discharge or area parameter for considering a flood event. The topic is suitable for publication in HESS but major revisions would be necessary in my opinion before the paper be published. General: - The paper lacks a discussion on the consistency of the choices to be made for designing the extremity indices (determination of Qs and the threshold Qs/Qma). There is almost no discussion about this point which constitutes the basis of the whole approach. Also, it seems from the discussion/conclusion section that the main difference between this work and previous other ones upon which the present study builds relies on the choice of the threshold selected for discharge: 1) this emphasizes even more the importance of strengthening the discussion on criteria for choosing the best suited Qs/Qma threshold, 2) it questions the value-added of including an area parameter in the approach (the authors themselves state that extremity indices are not very sensitive in changes in the area parameter: if so, then this approach is very similar to previous ones?).

**Authors:** Thank you for your comment. We changed discharge limits and use return periods for determining the event sets. Also, the subcatchment area is replaced by the length of river sections of certain order.

**Referee:** Conducting a more detailed study on the determinism of the occurrence of flood events seems important in order to relate the extremity indices defined to more concrete or practical hydrological/hydrometerological processes (in this sense it is surprising that the role of ground water is never even mentioned), it should be addressed here and would certainly constitute the value-added to other previous studies such as those of Uhlemann et al., etc. As a first step, the authors could try to relate the interannual variability and trends of extremity indices to some climate indices for instance.

**Authors:** The climatology of extreme flood events is just the first step of the research. The distribution of extreme flood events can be compared e.g. to some drought indices. This comparison and also the role of the antecedent wetness conditions are briefly mentioned within the discussion section.

**Referee:** Specific comments: - Title: Something like "large spatial extent floods" or "extensive floods" (as used in the introduction) could be included in the title to be more specific as it is an important aspect, and would prevent from using "trans-basin" which indeed could be misleading?

**Authors:** We included the term "extensive floods" in the title.

**Referee:** P.4, line 8: I don't get why only the downstream sub-catchment area is considered when an upstream gauging station is available. The downstream station is still representative of flow occurring over the whole upstream area anyway unless the upstream part of flow is substracted.

**Authors:** You are right. One possibility would be to substract the upstream part of flow. On the other hand, the actual discharge at certain station cannot be ignored, as it is related to severity of flood in that place. However, on the basis of other comments, we used the length of river sections of certain order instead of subcatchment areas.

**Referee:** P.4-5, "Methods": I do not recommend using the word "significant" in this context, as this does not refer to any statistical meaning here. I think the rationale for using this method to determine a time series of "significant" discharge values lacks explanation.

**Authors:** We agree and remove the word from the text. Moreover, when using return periods as discharge limits, the identification of what we called "significant" mean daily discharges is released.

**Referee:** As well, I am concerned by the approach for determining a significant flood event: the choice of the length of the time window needs a little more explanation. As is, it looks like the method suffers from a lack of either statistical or deterministic basis, and the definition of a flood event seems to be too much data- and operator-dependent and notenough transposable (see for instance "After analyzing all of the data series, we chose a time window [...]"). The fact that the time window had to be extended for one river, or that an additional rule had to be included to prevent merging events that have different atmospheric origins is also problematic: is an automatic split of flood events in two parts when they are separated by 5 days enough to conclude to different atmospheric causes ?

**Authors:** The time window of 10/12 days was used because of time of propagation of flood waves downstream. For example, during the event of March 1979, it took 11 days from first detected peak to the 10-year discharge value observed at Ketzin station on Havel river. To simplify it, we use single time window of 12 days before and after the observed 10-year discharge. If two or more flood waves occur, it is clearly visible in time series - two peaks greater than 10-year discharge appear at several stations and the distance between these peaks at each station is at least 5 days. E.g. it was the case of August 2002 flood, when two flood waves were detected, which corresponds to other studies about this event.

**Referee:** P.5, line 19: does the separation date between the cold and warm halves of the year also hold for other regions than the Czech Republic?

**Authors:** We decided for this separation mainly because of flood in 1998, which started on October 29. This flood clearly belongs to cold half-year floods due to its meteorological causes. Nevertheless, we finally used more classic division when the cold half-year start in November and this particular flood can be listed in cold half-year as the mean day of the event (derived according to Black and Werritty, 1997) occur in November.

[revised manuscript text omitted]

$$\text{\sout{A}L} = \sum_{i=1}^{k} l\text{\sout{a}}_i \tag{2\sout{1}}$$

where $\underline{l}\text{\sout{a}}_i$ denotes the  length of the river segment belonging to  one of $k$  stations where $Q_{s\underline{2}}$ is detected. The considered part of the river network upstream the station $i$ consists of individual river segments of a certain order. Strahler's stream ordering method is used (Strahler, 1957) when the first order is assigned to a headstream. Stream orders increase when two river segments of the same order meet. This method is dependent on the chosen layer of the river network. In this study, we use European catchments and Rivers network system of the European Environment Agency (EEA). However, only rivers of certain orders are included in the river length $l_i$. If a station is located on a stream of the fourth order, we consider only this particular river segment upstream the station. In the case of the fifth and sixth orders, also river segments of one lower order are counted. Two lower orders are considered when station is located on the stream of the seventh and eighth order. ~~The 80 largest floods are further examined. First, they are sorted based on whether they occurred during the colder or the warmer half of the year; the decisive day for classification is the first day with $Q_s$. The colder half-year is set between 16 October and 15 April because there is evidence from the Czech Republic of a relatively sharp interface in terms of flood occurrence in mid-April (Müller et al., 2015).~~

Both the spatial extent of floods and the aspect of the discharge magnitudes must be incorporated into an extremity index for evaluating extreme flood events. To demonstrate the role of the threshold of the considered maximum discharges, we defined  three index variants with differences in  discharge limits and applied them to the  identified flood events.

Generally, the index is derived from $A\underline{L}$ by multiplying $l\text{\sout{a}}_i$  by normalized peak discharges.  The basic variant consider all of the $Q_{sp}$ values normalized by the respective exact value of the 2-year return period $Q_{ma2}$:

$$E_{\text{\sout{a}}2} = \sum_{i=1}^{k} \left( \frac{Q_{spi}}{Q_{ma2i}} \, \text{\sout{a}}l_i \right). \tag{3\sout{2}}$$

[revised manuscript text omitted]

**Figure 3: The 30 largest flood events in the study area from 1951 to 2013 according to $E_2$ and the corresponding events according to $E_{10}$. Missing bars indicate events which are not included in the set of 30 largest floods compiled by $E_{10}$. Contributions of individual river basins to the index value are distinguished by color. Red dots indicate warm half-year floods.**

[Figure]

**Figure 4: The occurrence of discharges equal to or greater than 2, 5 and 10-year flood at individual stations during each of the 30 maximum floods according to $E_2$ index. The basins are indicated at the top of the chart; the stations are arranged according to their position downstream.**

[Figure]

[Figure]

**Figure 54: Seasonal distribution of 30 maximum floods according to $E_{r2}$ and $E_{1.5r10}$ indices. The number of extreme floods in individual months N is depicted by shading; the mixed color indicates overlapping data. The signs represent mean calendar days of the events; the distance of the sign from the center of the diagram reflects the flood extremity given by the value of $E_{r2}$ [10³] and $E_{1.5r10}$ [5·10²].**

[Figure]

[Figure]

**Figure 56: Interannual variability of 30 major floods according to $E_{r2}$ and $E_{1.5r10}$.  The symbols represent the extremity of cold and warm half-year floods  with respect to $E_{r2}$ [$10^3$] and $E_{1.5r10}$ [$5 \cdot 10^2$]; Lines depict  linear trends and relative cumulative values of the flood extremity.**

[Figure]

[Figure]

**Figure 67: Spatial distribution of 30 maximum floods according to $E_{r2}$ (a, b) and $E_{1.5r10}$ (c, d). The numbers of cold (a, c) and warm (b, d) half-year floods identified in individual gauge stations duringsubcatchments during 1950–20101951-2013 are depicted by circle sizeshading.**